# Genetic Mapping of a Candidate Gene *ClIS* Controlling Intermittent Stripe Rind in Watermelon

Yinping Wang [†], Shixiang Duan [†], Qishuai Kang, Dongming Liu, Sen Yang (iD), Huanhuan Niu, Huayu Zhu, Shouru Sun, Jianbin Hu, Junling Dou * and Luming Yang *

College of Horticulture, Henan Agricultural University, Zhengzhou 450002, China
* Correspondence: doujunling@henau.edu.cn (J.D.); lumingyang@henau.edu.cn (L.Y.)
† These authors contributed equally to this work.

**Abstract:** Rind pattern is one of the most important appearance qualities of watermelon, and the mining of different genes controlling rind pattern can enrich the variety of consumer choices. In this study, a unique intermittent rind stripe was identified in the inbred watermelon line WT20. The WT20 was crossed with a green stripe inbred line, WCZ, to construct $F_2$ and $BC_1$ segregating populations and to analyze the genetic characterization of watermelon stripe. Genetic analysis showed that the intermittent stripe was a qualitative trait and controlled by a single dominant gene, *ClIS*. Fine mapping based on linkage analysis showed that the *ClIS* gene was located on the 160 Kb regions between 25.92 Mb and 26.08 Mb on watermelon chromosome 6. Furthermore, another inbred watermelon line with intermittent stripe, FG, was re-sequenced and aligned on the region of 160 Kb. Interestingly, only two SNP variants (T/C, A/T) were present in both WT20 and FG inbred lines at the same time. The two SNPs are located in 25,961,768 bp (T/C) and 25,961,773 bp (A/T) of watermelon chromosome 6, which is located in the promoter region of *Cla019202*. We speculate that *Cla019202* is the candidate gene of *ClIS* which controls the intermittent stripe in watermelon. In a previous study, the candidate gene *ClGS* was proved to control dark green stripe in watermelon. According to the verification of the two genes *ClIS* and *ClGS* in 75 watermelon germplasm resources, we further speculate that the *ClGS* gene may regulate the color of watermelon stripe, while the *ClIS* gene regulates the continuity of watermelon stripe. The study provides a good entry point for studying the formation of watermelon rind patterns, as well as providing foundation insights into the breeding of special appearance quality in watermelon.

**Keywords:** watermelon; intermittent stripe; mapping; rind formation





## 1. Introduction

The appearance quality of fruit is a critical attribute that heavily influences consumer preference. A distinctive and novel rind pattern can greatly enhance the appeal of a fruit, and therefore have a significant impact on consumer choice. Thus, consumer habits play a crucial role in shaping plant evolution and domestication [1]. As an important appearance trait, fruit rind pattern has many kinds of variation, especially in horticultural crops. In tomato, the fruit peels are distinguished as red, dark green, light green, green shoulders, dark green stripe and so on. It has been reported that many different genes regulate tomato peel color and pattern. *APRR2-Like* and *TKN2* genes regulate the formation of dark green fruit by increasing the accumulation of chlorophyll content [2,3]. *TKN4* and *TKN2* regulate gradient expression of *GLK2* and *APRR2-Like* to establish chloroplast developmental gradient from fruit shoulder to hilum during tomato fruit development, which ultimately leads to the formation of green shoulder tomato fruit [3,4]. Furthermore, methylation of *TAGL1* promoter has been identified as regulating the formation of green stripes in tomato peel [5]. In cucumber, transcription factor *MYB36* has been identified as regulating the formation of yellow-green fruit [6], while two genes, *Csa7G05143* and *Csa6G133820,* were

reported to associate with light-green rind [7,8], and the gene *Csa3G904140* was associated with immature rind color in cucumber [9]. The gene *Csa1G005490*, encoding polygalacturonase, was identified as regulatingan irregular striped pattern in cucumber [10]. In apple, the transcription factor *MYB10* was identified as regulating a red stripe in fruit rind [11], and many other transcription factors (*MdMYB1a*, *MdMYB1*, *MdbHLH3-1*, *MdbHLH33-1*, *MdUFGT2-1*, and *MdUFGT4*) which are involved in anthocyanin synthesis can also affect rind color [12]. In addition, many other genes have also been identified controlling fruit rinds in pepper [13], pear [14], wax gourd [15], sweet cherry [16] and so on.

A member of the Cucurbitaceae, watermelon (*Citrullus lanatus*) is grown widely and very popular with consumers all over the word. It has 11 chromosomes (2n = 2X = 22) with an approximately 425 Mb reference genome size. Watermelon reference genome '97103' v1 was sequenced and released in 2013 [17]. Subsequently, the 'Charleston Gray' reference genome and '97103' v2 have also been released [18]. All these make it possible to study the inheritance pattern and functional genes of some important agronomic traits in watermelon. As an important appearance quality, watermelon rinds are rich in diversity. According to the background color, watermelon rinds can be distinguished as yellow, dark green, light green, white and so on. According to the stripe patterns, watermelon rinds can be distinguished as striped, netted, intermittent, mottled and so on [19–21]. In addition, there are some other special rind patterns, such as yellow spots, yellow belly, and uneven rind [21–23]. Different kinds of rind pattern have always been the research hotspot of watermelon breeders.

In pervious studies, the research on watermelon rinds has focused on inheritance and preliminary mapping. Inheritance analysis has shown that light green rind color is controlled by a single gene *g*, with light green exhibiting recessive behavior compared to the dominant dark green rind color [21,22,24]. However, it has been reported that two loci together regulate dark green and light green rind patterns [25]. Yellow rind color has been confirmed to be controlled by a dominant gene [20,26]. For watermelon stripes, a series of alleles have been identified to control different stripe patterns, and the dominance relationship of solid medium/dark green (*G*) > wide stripe ($g^W$) > medium stripe ($g^M$) > narrow stripe ($g^N$) > light green (*g*) has also been identified [27]. In addition, some other special rind patterns have also been investigated. For example, the intermittent stripe pattern has been confirmed to be controlled by a single recessive gene [20]. A single dominant gene was identified as regulating watermelon yellow belly [20]. The 'moon and star' pattern, with different kinds of yellow spots on the fruit rind, has been shown to be regulated by two genes [23].

With the reference genome sequences available, a total of 32,869 SSR markers distributed throughout watermelon whole genome were developed according to '97103' genome [28], which makes it easy to carry out fine mapping of the functional genes and to study the regulatory mechanisms. To date, few genes and loci controlling watermelon rind traits have been reported. It has been reported that the gene *RS8.1*, located on chromosome 8, was identified to control dark green rind through the construction of a genetic map using $BC_1$ generations crossed by watermelons with dark green rind and light green rind [29]. Furthermore, the *ClCGMenG* gene on chromosome 8 encoding a methyltransferase was identified to be responsible for watermelon dark green rind [30]. Additionally, a 3-bp insertion/deletion in the *ClGS* gene may be responsible for dark green stripes [31]. Yellow rind is regulated by a locus on the forefront of chromosome 4 [26]. For the 'moon and star' trait, two gene loci distributed on chromosomes 1 and 8 were identified as playing the decisive role [23]. In addition, studies have shown that several loci on chromosome 6 were identified to be related to watermelon rind stripes [32–35]. The 611.78 kb region on chromosome 6, which consistsof four discrete haploblocks and continuous recombination suppression, was identified as regulating watermelon stripe patterns [35]. A similar result was also found in genome-wide association analysis of 414 different watermelon accessions [36]. However, the research on fine mapping and cloning of some special watermelon rind patterns, especially intermittent stripes, have not yet been reported.

In our study, an intermittent stripe watermelon inbred line WT20 and a green stripe watermelon inbred line WCZ were crossed to construct an $F_2$ population. The inheritance analysis showed that the intermittent stripe was controlled by a single dominant gene, which was named *ClIS* (*Citrullus lanatus Intermittent Stripe*). Further research has shown that *Cla019202* may be the candidate gene for intermittent stripes, and that variation in the *Cla019202* promoter region may be responsible for the continuity of watermelon stripes. This study provides valuable insights into the regulatory mechanisms involved in watermelon rind stripe formation and organ development, and it holds great potential to assist breeders in marker-assisted selection, manipulation of rind pattern, and improvement of fruit quality during watermelon breeding.

## 2. Materials and Methods

### 2.1. Plant Materials and Phenotypic Measurement

In our study, the fruit rind of the watermelon accession WT20 displayed an intermittent stripe pattern, which could be observed 10 days after pollination (DAP). On the other hand, the watermelon accession WCZ was characterized by a green stripe pattern, which became apparent during ovary development (Figure 1). Both accessions were homozygous inbred lines which had been self-pollinated for more than five generations. WT20 ($P_1$) and WCZ ($P_2$) were used to construct a segregation population of watermelon stripe patterns for gene mapping and genetic analysis. WT20 was crossed with WCZ to obtain $F_1$, and the $F_2$ population was obtained by self-crossing of $F_1$. The populations $BC_1P_1$ ($F_1 \times P_1$) and $BC_1P_2$ ($F_1 \times P_2$) were obtained through the backcrossing of $F_1$ with each parent.

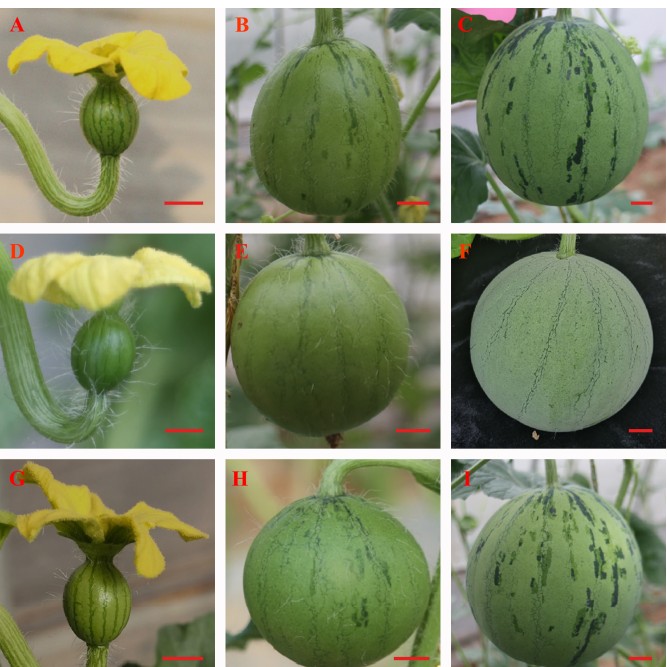

**Figure 1.** The phenotypes of watermelon inbred line WT20, WCZ, and $F_1$. (**A**) The ovary period of WT20. (**B**) 10 days after pollination of WT20. (**C**) 30 days after pollination of WT20. (**D**) The ovary period of WCZ. (**E**) 10 days after pollination of WCZ. (**F**) 30 days after pollination of WCZ. (**G**) The ovary period of $F_1$. (**H**) 10 days after pollination of $F_1$. (**I**) 30 days after pollination of $F_1$. Bar = 2 cm.

For genetic analysis, the $F_2$ population was grown and investigated in three experiments conducted in spring 2020, autumn 2020, and spring 2021 with 130, 217, and 849 $F_2$ individuals, respectively, while 59 $BC_1P_1$ individuals and 59 $BC_1P_2$ individuals were grown and investigated in spring 2020 only. The parents with 10 individuals and $F_1$ with 10 individuals were included in all experiments. The phenotyping of stripe pattern of each

individuals was investigated and recorded two times at 10 DAP and 30 DAP (mature period). The $\chi^2$ test was used to calculate the segregation ratio in $F_2$ population and backcross population (Table S1). All the individuals in this study were grown in the greenhouses in the Science and Education Park of Henan Agricultural University, Zhengzhou, China.

## 2.2. Marker Development and Fine Mapping

The whole genome sequence of watermelon '97103' was released to the Cucurbit Genomics Database (http://www.icugi.org. accessed on 1 January 2020) in 2013 [17]; a total of 32, 869 SSR markers distributed in the watermelon whole genome were developed according to the '97103' reference genome [28]. In order to confirm the candidate region of watermelon stripe pattern, we screened the SSR markers in the watermelon whole genome to analyze the polymorphism between the parents WT20 and WCZ. Then, in the $F_2$ population in autumn 2020, we constructed two DNA pools: the intermittent stripe pool (I-pool) and the green stripe pool (G-pool), by mixing an equal amount of DNA from 20 intermittent stripe individuals and 20 green stripe individuals. The SSR markers which were polymorphic between WT20 and WCZ were used to screen the polymorphisms between two DNA pools. Finally, the polymorphic markers were used for linkage analysis through the $F_2$ population of spring 2021 (849 individuals) and autumn2020 (217 individuals), which totaled 1066 individuals (Table S1). The primary mapping was based on SSR markers as above, and then Indel markers and dCAPS markers were developed during the interval of primary mapping according to the re-sequencing data for fine mapping in the $F_2$ population. The Indel markers and dCAPS markers were designed based on the Primer3 software (http://bioinfo.ut.ee/primer3-0.4.0/. accessed on 12 January 2022) and dCAPS Finder 2.0 [37], respectively. The PCR reaction system, reaction procedure, restriction enzyme digestion reactions, and subsequent gel electrophoresis were conducted as described by Dou et al. [38]. The key markers used for linkage analysis in this study are listed in Table S2.

## 2.3. Prediction and Verification of Candidate Gene

The predicted gene in the candidate region was retrieved from the watermelon reference genome '97103' (http://cucurbitgenomics.org/. accessed on 12 January 2022). The gene-specific primers were designed according to the DNA and CDS sequence of the candidate gene. The DNA and CDS sequence of the candidate gene were amplified from the ovary of two parents WT20 and WCZ, and then PCR products were sequenced and aligned between the '97103' reference genome and two parents for detecting sequence differences. The function of the candidate gene was predicted based on NCBI (https://blast.ncbi.nlm.nih.gov/Blast.cgi. accessed on 12 January 2022) and the Cucurbitaceae Genome Database (http://cucurbitgenomics.org/. accessed on 12 January 2022). Furthermore, a watermelon accession which has the intermittent stripe pattern, FG, was obtained from Henan Yuyi Seed Co., Ltd., Zhengzhou, China. The stripe pattern of FG was the same as WT20, while other traits such as seed color and seed size were different between WT20 and FG. To verify the candidate gene of intermittent stripe watermelon, FG was re-sequenced for further testing.

In a previous study, the candidate gene *ClGS* (Cla019205) was provento control dark green stripe, and an Indel marker was developed according to the sequence of *ClGS* [31]. In this study, 75 watermelon germplasm resources (Table S3), including 41 darkgreen stripe watermelons, 32 green stripe watermelons and 2 intermittent stripe watermelons were used to test the linkage of candidate gene and phenotype. The sequences of intermittent stripe gene *ClIS* from watermelon germplasm can only be amplified and sequenced because of two close SNPs existing in the exon regions.

## 2.4. RNA Extraction, cDNA Synthesis, and qRT-PCR Analysis

In order to examine the expression of the intermittent stripe gene *ClIS* and the dark green stripe gene *ClGS*, watermelon rind displaying three kinds of stripe pattern (in-

termittent stripe WT20, green stripe WCZ, and dark green stripe WT2) was isolated at 10 DAP, and RNA was extracted for investigating the expression pattern of candidate gene. Total RNA was extracted using the plant RNA purification kit (Huayueyang, China) according to the protocol of the manufacturer. The quantity and quality of RNA samples were determined and assayed using NanoDrop 2000 (Thermo) and 1.5% agarose gel, respectively. The cDNA was synthesized using reverse transcriptase M-MLV (RNase H-) based on the instructions of the manufacturer (Takara, Japan). The primers for real-time PCR were designed for amplifying products of 100–250 bp using the Primer 5.0 online software. The *Actin* primer [39] and genes primers are listed in Table S4. The qRT-PCR reaction was carried out as previously reported [38]. The experiments were repeated three times, both biologically and technically, and the relative expression pattern was performed and calculated using the $2^{-\triangle\triangle CT}$ method [40].

## 3. Results

### 3.1. Inheritance Features of Intermittent Stripe in Watermelon Rind

The intermittent stripe phenotypein the inbred line WT20 can be easily distinguished at 10 DAP, and becomes more and more obvious during later development of the fruit. Conversely, the inbred line WCZ shows a green stripe pattern throughout the fruit development period, from ovary to fruit maturity (Figure 1). Both inbred lines had been manually self-pollinated for at least five generations and were genetically stable. In order to study the genetic regulation of the intermittent stripe, WT20 ($P_1$) and WCZ ($P_2$) were used as parents to construct genetically segregated populations. All the fruits of $F_1$ individuals displayed an intermittent stripe, suggesting that watermelon intermittent stripe is dominant in this study (Figure 1). $F_2$ and $BC_1$ segregated populations were constructed through the cross of WT20 and WCZ. The segregation ratio of the stripe pattern among different populations grown in three replicates is presented in Table S1. In spring 2020, there were 100 intermittent stripe individuals and 30 green stripe individuals among 130 $F_2$ plants. All 59 $BC_1P_1$ individuals showed an intermittent stripe, while out of 59 $BC_1P_2$ individuals, 33 and 26 showed an intermittent stripe and green stripe, respectively. In autumn 2020, 164 and 53 individuals showed intermittent stripes and green stripes, respectively, out of 217 $F_2$ individuals. In spring 2021, 849 $F_2$ individuals were grown, of which 622 were intermittent stripe individuals and 227 were green individuals. The χ2 test showed that the segregation ratios in the $F_2$ population and the $BC_1$ population were in line with 3:1 and 1:1, respectively (Table S1), indicating that watermelon intermittent stripe is a qualitative trait and regulated by a single dominant gene. We named this gene *ClIS* (*Citrullus lanatus Intermittent Stripe*).

### 3.2. Fine Mapping of the ClIS Gene

In our previous study, a total of 1,257 SSR genome-wide markers were developed [28]. Of these markers, 532 showed good polymorphism between the two parental lines, WT20 and WCZ. Two pools (I-pool and G-pool) mixed an equal amount of DNA from 20 intermittent stripe individuals, and 20 green stripe individuals were used for preliminarily mapping the locus of *ClIS*. Among the 532 SSR markers, nine distributed on watermelon chromosome 6 presented good polymorphism between the two pools (Table S2). According to the location of nine SSR markers, the *ClIS* gene locus was located on the end of the watermelon chromosome 6 (Figure 2A). To check these results and map the candidate gene of watermelon intermittent stripe, 217 $F_2$ individuals from autumn 2020 were used for polymorphic analysis using the nine polymorphism markers. The *ClIS* gene was mapped to 340 Kb genetic distances between ClSSR18071 and ClSSR18124 on watermelon chromosome 6 (Figure 2A).

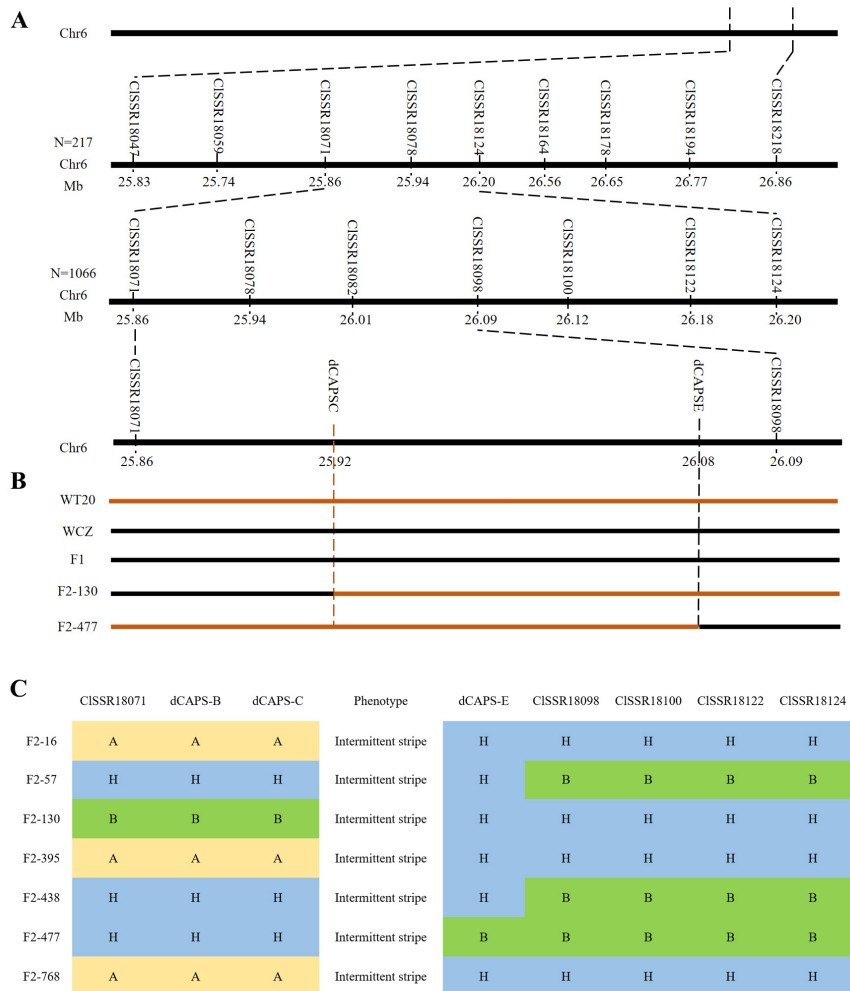

**Figure 2.** Linkage analysis based on the $F_2$ population.The candidate gene was located between dCAPS-C and dCAPS-E markers on chromosome 6. (**A**) The linkage analysis of the $F_2$ population. (**B,C**) The genotype of recombinant individuals: A is responsible for dominant homozygosity; B is responsible for recessive homozygosity; H is responsible for heterozygosity.

To further narrow the candidate region of *ClIS* gene, an additional five SSR markers with good polymorphism between markers ClSSR18071 and ClSSR18124 were developed (Table S2). These markers were used to perform a more detailed linkage analysis and fine mapping using 1066 $F_2$ individuals, which contained 217 individuals from autumn 2020 and 849 individuals from spring2021. The linkage analysis showed that the *ClIS* gene was mapped to 230 Kb genetic distance between markers ClSSR18071 and ClSSR18098 on watermelon chromosome 6 (Figure 2A). To obtain more markers for fine mapping, two parents of WT20 and WCZ were re-sequenced and analyzed by aligning with the reference genomes. Based on the re-sequenced data between the two parents, two dCAPS markers and one Indel marker between markers ClSSR18071 and ClSSR18098 were developed and used for polymorphic analysis in the recombinant individuals. The candidate gene *ClIS* was finally located at the interval between markers dCAPS-C and dCAPS-E (Figure 2B). Using the physical position of markers dCAPS-C and dCAPS-E, the *ClIS* gene was finally fine mapped to the 160 Kb region between 25.92 Mb and 26.08 Mb on watermelon chromosome 6 (Figure 2).

### 3.3. Sequence Alignment and Gene Identification

According to the annotation database of the watermelon reference genome '97103' (http://cucurbitgenomics.org/. accessed on 12 January 2022), 18 functional genes (from Cla019198 to Cla019215) have been annotated in the 160 Kb candidate region (Figure 3). The sequences

of these 18 functional genes were amplified and blasted. The results showed that there were no base differences in exons among WT20, WCZ, and the reference genome. Furthermore, another intermittent stripe watermelon accession, FG, was re-sequenced for further analysis of the variation in the 160 Kb candidate region. We blasted the sequence differences in the candidate region between WT20 and FG because they share the same stripe pattern. Interestingly, only two SNP variants (T/C, A/T) were present in both the WT20 and FG inbred lines (Figure 4A). The two SNPs were at 25,961,768 bp (T/C) and 25,961,773 bp (A/T) of watermelon chromosome 6. Furthermore, the two SNPs were in the promoter region of Cla019202: one was at the 1538th base (T/C) in the front of the start codon, and another was at the 1533rd base (A/T) in the front of the start codon. Analysis of promoter action elements (http://bioinformatics.psb.ugent.be/. accessed on 12 January 2022) showed that there was an additional CTCC-binding element in the promoter region of the Cla019202 gene (Figure 4A). However, the two SNPs were also present in the green stripe watermelon WCZ. In a previous study, the candidate gene *ClGS* (Cla019205) was proved to control watermelon dark green stripe, and a 3 bp insertion was identified in watermelons with nodark green stripe [31]. In order to further verify the candidate gene of dark green stripe pattern and intermittent stripe pattern, 75 watermelon germplasm resources selected from a previous study [36] were used to test the linkage between gene and phenotype (Table S3). The results showed that there were no SNPs of the promoter of Cla019202 presented in 41 dark green stripe watermelon accessions, while 3 bp insertions in *ClGS* gene were also not present, which was the same as in the previous study [31]. For the 32 green stripe watermelon accessions, there were two SNP variations in the promoter of Cla019202, and a 3 bp insertion in the *ClGS* gene was also present. For the two intermittent stripe watermelon accessions, there were two SNP variations in the promoter of Cla019202, and the 3 bp insertion in the *ClGS* gene was not presented (Figure 4B). We speculate that the genes Cla019202 (*ClIS*) and Cla019205 (*ClGS*) together regulate three stripe patterns of watermelon rind.

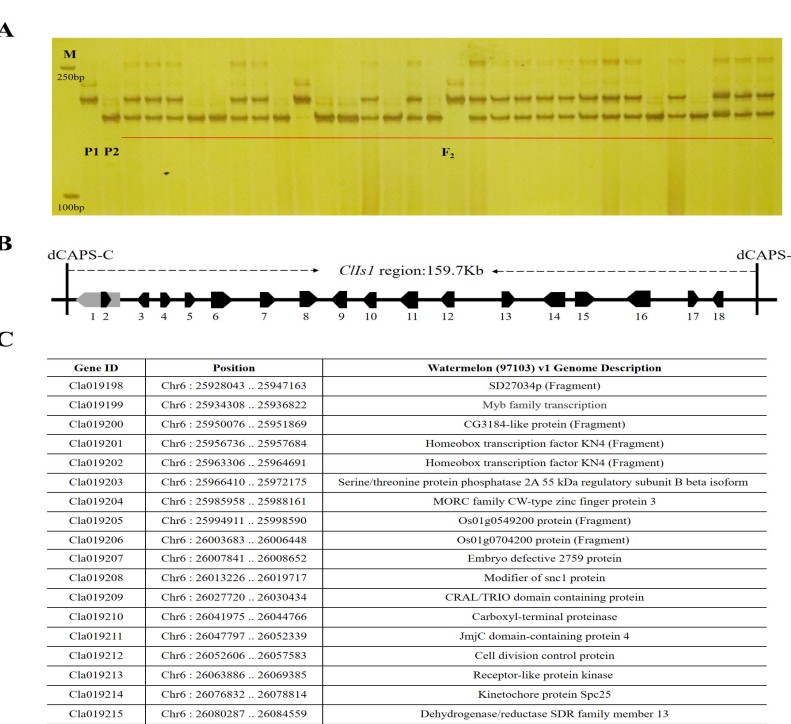

**Figure 3.** The functional genes between dCAPS-C and dCAPS-E markers. (**A**) The F$_2$ population was co-segregated in the candidate interval. (**B**) 18 function genes were present in candidate interval. (**C**) Details information of the 18 functional genes.

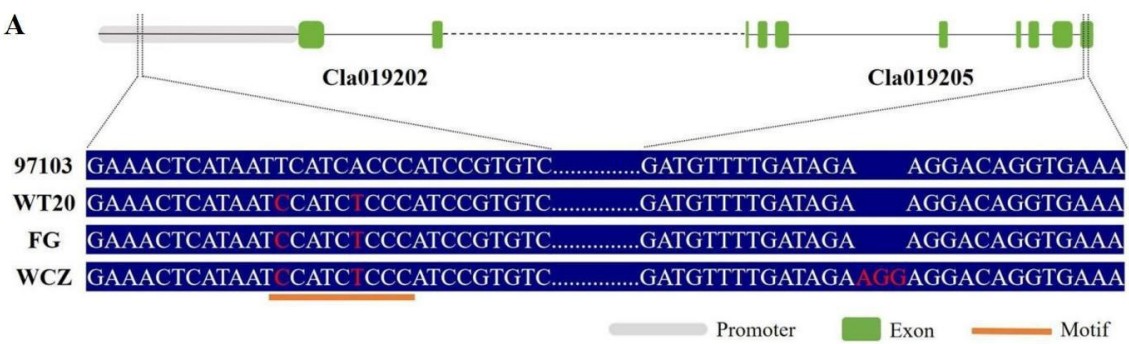

**Figure 4.** Co-segregation verification of the *ClIS* and *ClGS* genes. (**A**) Sequence alignment of the *ClIS* and *ClGS* genes among different stripe patterns; 97103, dark green stripe; WT20 and FG, intermittent stripe; WCZ, green stripe. (**B**) The verification of *ClIS* and *ClGS* genes among 75 watermelon germplasm resources.

### 3.4. Expression Analysis of ClIS and ClGS Genes

In order to analyze the expression pattern of *ClIS* (*Cla019202*) and *ClGS* (*Cla019205*) genes in differently striped watermelon, the rind tissues at 10 DAP from three watermelons (intermittent stripe watermelon WT20, dark green stripe watermelon WT2, green stripe watermelon WCZ) were isolated and RNA was extracted for investigating the expression pattern. The specific primers of *ClIS*, *ClGS*, and the reference gene *Actin* are listed in Table S4. For the *ClIS* gene, there was no significant difference in expression level between WT20 and WCZ, but the expression level in WT20 and WCZ were higher than it was in WT2. For the *ClGS* gene, there was no significant difference in expression level between WT2 and WT20, but the expression levels in WT2 and WT20 were significantly lower than in WCZ (Figure 5). The stripe pattern could be easily distinguished at 10 DAP, which corresponded to the expression pattern of two genes. We speculate that different combinations of *ClIS* and *ClGS* determined different gene expression level, and then affected the formation of different watermelon stripe patterns.

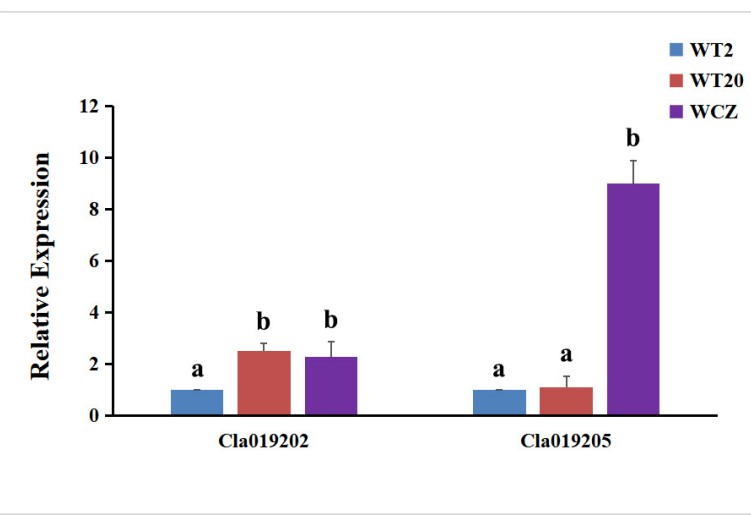

**Figure 5.** Expression analyses of *ClIS* and *ClGS* genes in fruit of 10 DAP. a and b represent significant differences.

*3.5. Formation of Watermelon Stripe Patterns*

In a previous study, the *ClGS* gene was shown to regulate the formation of dark green stripes [31]. The expression level of the *ClGS* gene was also the same as the previous study, so we determined that the *ClGS* gene regulates watermelon dark green stripes. For the intermittent stripe watermelon, the variation in the promoter region of *ClIS* can result in changes of expression level, and then affect the formation of the intermittent stripe pattern (Figure 6). According to the verification of co-segregation and expression level of the two genes (Figures 4, 5), we think that the *ClGS* gene regulates the color of the watermelon stripe, while the *ClIS* gene regulated the continuity of the watermelon stripe (Figure 6).

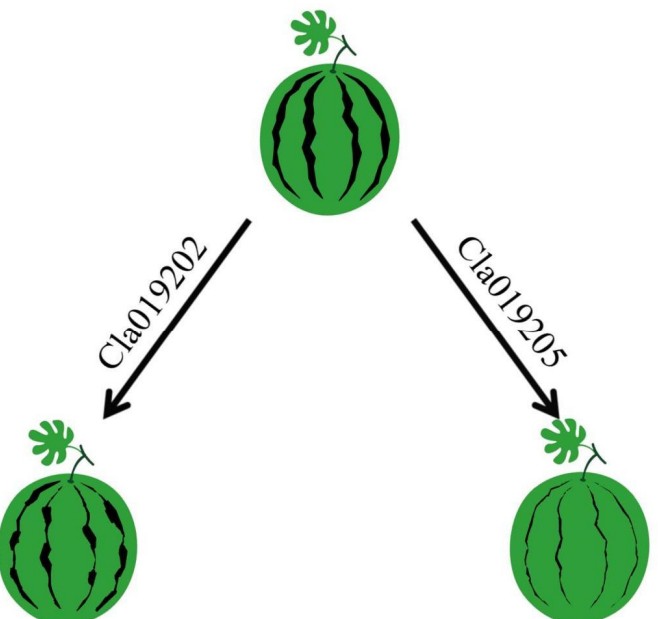

**Figure 6.** The predictive regulatory model of watermelon stripe pattern.

## 4. Discussion

Watermelon is one of the most popular and widely cultivated fruits all over the world. As an important appearance trait, there have been reports on the study of watermelon rind pattern. However, the research mainly focused on common traditional

rind patterns, such as dark green rind color background regulated by the *ClCGMenG* gene [30], dark green stripe regulated by the *ClGS* gene [31], different stripe patterns regulated by four discrete haploblocks of chromosome 6 [35], and yellow rind color regulated by a locus on the forefront of chromosome 4 [26]. There have been few reports on unique watermelon rind patterns. In the present study, an inbred watermelon line with intermittently striped rind, WT20, which had been manually self-pollinated for at least five generations, was collected and preserved (Figure 1). WT20 was the ideal material for researching watermelon intermittent stripe pattern in view of its stable inheritance without trait segregation during cultivation. The $F_2$ populations were constructed by crossing between WT20 and WCZ (green stripe). Segregation in $F_2$ populations exhibited the ratio of 3 intermittent stripes: 1 green stripe (Table S1), which indicated that watermelon intermittent stripe is a qualitative trait that is regulated by a single dominant gene.

With the release of the watermelon reference genome, gene mapping and cloning have accelerated using the technology of NGS (Next Generation Sequencing), and the efficiency of genotyping has also improved. NGS-assisted BSA provides an effective and simple method to identify molecular markers linked to target genes/QTLs because it is less laborious, much cheaper and has no population size limitation for genotyping work. It can be performed by obtaining a sufficient number of plants with extreme traits in a specific population [41]. Using the combination of BSA-seq technology and SSR markers distributed throughout the watermelon genome, several genes in watermelon have been mapped and cloned, such as the *ClFS1* gene controlling fruit shape [42], the *ClCGMenG* gene controlling rind color [30], the *Cldf* gene controlling plant dwarfism [43,44], the *ClERF4* gene controlling rind hardness [45], and the *ClTFL1* gene controlling branchlessness [38]. This study utilized BSA-seq technology along with linkage analysis to map the *ClIS* gene.The candidate gene was delimited in a 160 Kb region between 25.92 Mb and 26.08 Mb on chromosome 6 (Figure 2), and 18 functional genes (from Cla019198 to Cla019215) were annotated in the 160 Kb candidate region (Figure 3). Additionally, an inbred line of intermittent stripe watermelon (FG) was collected and re-sequenced. The 18 genes were aligned and the differences in the sequences among WT20, WCZ, FG and 97103 (reference genome, dark green stripe) were analyzed. However, no differences were found among these 18 genes in the four watermelon accessions. Previous research has identified the candidate gene *ClGS* (Cla019205) as controlling watermelon dark green stripes and found a 3 bp insertion in watermelons without dark green stripes [31]. Our further analysis found that the 3 bp insertion in the *ClGS* gene was only presented in the green stripe watermelon WCZ. Interestingly, only two SNP variants (T/C, A/T), located on the promoter region of *Cla019202*, were present in WT20, FG, and WCZ inbred lines, which was different with dark green stripe watermelon 97103 (Figure 4A). Linkage analysis among 75 watermelon germplasm resources showed that 32 green striped watermelons presented variation in two SNPs in the *Cla019202* gene and a 3 bp insertion in the *ClGS* gene, 41 dark green stripe watermelons showed no variation in the two genes, and two intermittent stripe watermelons only present in two SNPs variation were in the *Cla019202* gene (Figure 4B). So we speculate that *Cla019202* is the candidate gene of *ClIS* which regulated watermelon intermittent stripe rind. Additionally, the *ClGS* gene regulates the color of watermelon stripe, while the *ClIS* gene regulated the continuity of watermelon stripe (Figure 6).

It has been reported that the accumulation of pigment plays an important role in fruit rind. Many genes related to the biosynthesis of different pigments have been mapped and cloned. The *CCD4b* gene (carotenoid cleavage dioxygenase 4b) affects the color of *Citrus* peel by regulating the production of β-citraurinene [46]. Many *MYB* transcription factors have been identified as regulating pigment biosynthesis. *CsMYB36* transcription factor is responsible for the formation of cucumber yellow-green rind [6]. The methylation of *MYB10* transcription factors regulates the formation of fruit rind in apples and pears [11,14]. In addition, methylation in the promoter region of *TAGL1* regulates the accumulation of chloroplasts in different parts of immature tomato fruit rinds, resulting in the appearance



of green stripe traits in mature fruit [5]. In watermelons, there is some research on the regulation of watermelon rind stripe. GWAS analysis in 414 different watermelon accessions showed that the watermelon fruit stripe gene is linked to chromosome 6, and it has been predicted that *Cla97C06G126710* is the candidate gene of watermelon rind stripe [36]. Subsequent research showed that the 611.78 kb region on chromosome 6 which consisted of four discrete haploblocks and continuous recombination suppression was identified as regulating watermelon stripe pattern [35]. Wang et al. [31] identified that *ClGS* (*Cla019205*) was responsible for watermelon dark green stripe. However, there were no further researches on unique intermittent stripe rind patterns in watermelon.

In the present study, the *Cla019202* gene was identified to be the candidate gene for regulating watermelon intermittent stripe. Furthermore, a proposed model of watermelon rind stripe formation was established: the *ClGS* gene regulates the color of watermelon stripe, while the *ClIS* gene regulates the continuity of watermelon stripe. Genetic variants and gene mutation start from rare mutations with extremely low frequency, which appear in an individual from a specific population. The variants or mutation are then transformed into common ones during the evolutionary process through migration, selection, and genetic drift [38]. In watermelon, the genome speciation event occurred in 15–23 million years ago [17]. Wild watermelon originated in desert area, where itexhibited the phenotype of dark green or light-green rind stripes [36]. We speculate that the *ClIS* gene may have caused the natural variation in the long-term process of evolution and domestication, and that the variation was preserved through artificial selection. After many generations of selection and purification, the locus for intermittent stripe was preserved and inherited stably. In this study, it has been identified that the *ClIS* gene, combined with the *ClGS* gene, plays an important role in the development of watermelon rind patterns. However, the molecular mechanism of these two genes in regulating the formation of watermelon rind stripe development needs to be further studied. As a unique stripe trait in watermelon, intermittent stripe can be not only used as amorphological marker for cultivar identification, but can also provide the theoretical basis for the breeding and improvement of distinctive watermelon varieties. The cloning of watermelon stripe genes provides a good entry point for studying the development of watermelon rind, as well as providing foundation insights into breeding of different kinds of rind pattern in watermelon.

**Supplementary Materials:** The following supporting information can be downloaded at: https://www.mdpi.com/article/10.3390/horticulturae9020263/s1, Table S1. The segregation ratio of intermittent stripe and green stripe among different populations (WT20×WCZ). Table S2. The information of remaining important markers on chromosome 6 used to analysis the polymorphic. Table S3. The genotypes in 75 watermelon germplasm resources. Table S4. The information of qRT-PCR markers.

**Author Contributions:** Conceptualization and methodology, J.D. and L.Y.; software, S.D. and D.L.; validation, Y.W., S.Y. and H.Z.; formal analysis, S.Y.; investigation, D.L., H.N. and Q.K.; resources, J.D.; data curation, Y.W. and S.D.; writing—original draft preparation, J.D.; writing—review and editing, J.H. and L.Y.; visualization, H.Z. and S.S.; supervision, S.S.; project administration, J.D.; funding acquisition, L.Y. All authors have read and agreed to the published version of the manuscript.

**Funding:** This research was supported by the National Natural Science Foundation of China (32102389, 32172602), the Project for Scientific and Technological Activities of Overseas Students of Henan Province, the Zhongyuan Youth Talent Support Program (ZYQR201912161), the Funding of Joint Research on Agricultural Varieties Improvement of Henan Province (2022010503), the Major Science and Technology Project of Henan Province (221100110400), and the Science and Technology Innovation Fund of Henan Agricultural University (KJCX2021A14).

**Data Availability Statement:** All new research data were presented in this contribution.

**Conflicts of Interest:** The authors declare no conflict of interest.

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
