# Peer review of "Genetic Mapping of a Candidate Gene ClIS Controlling Intermittent Stripe Rind in Watermelon"

_horticulturae, doi:10.3390/horticulturae9020263_

Round 1

Reviewer 1 Report

The article is devoted to the study of the genetic control of the watermelon fruit pattern. The authors argue that breeding varieties with intermittent stripes is important for consumer product characteristics. I'm not entirely sure about this. It seems to me more important - what's inside plus productivity and sustainability. It may be interesting to consider other issues related to fruit color (which fruit is more visible or attractive to pestsб birds, etc.) I would analyze the literature and add an ecological bias to the introduction in terms of color diversity.

Everything else does not cause any remarks for me. The work was done to a good standard, in good faith.

Author Response

Response:

Thank you very much for your comments and suggestions.

1. Varieties diversity can increase consumers' choice. As a special peel type, intermittent stripe watermelon is not common. We hope to breed an intermittent strip watermelon variety with with good quality and strong resistance in the future to increase market diversity. So, we think that breeding varieties with intermittent stripes may be important for consumer product characteristics.

2. The key point of this manuscript is the gene mapping about watermelon intermittent strip. In this manuscript, we cloned the candidate gene which was responsible for watermelon intermittent strip. We had not focus on the productivity and sustainability about intermittent strip. Maybe we will pay attention to these traits in our next study. We also hope you can continue to pay attention to our follow-up researches. Thank you very much.

Reviewer 2 Report

In the manuscript entitled “Genetic mapping of a candidate gene ClIS controlling intermittent strip rind in watermelon” the authors generated a segregating F2 watermelon population by crossing an intermittent strip watermelon inbred line, WT20 and a green strip watermelon inbred line, WCZ and showed that the intermittent strip is controlled by the gene Citrullus lanatus Intermittent Strip (ClIS). Through genetic analysis the author’s demonstrated that watermelon intermittent strip is a qualitative trait under the regulation of ClIS, a single dominant gene. This finding of this study will be of particular importance to watermelon breeders. In selecting fruits, consumers are particularly attentive to quality and appearance. Features such as a unique rind pattern can be attractive to consumers. Watermelon is an important fruit which is grown and consume worldwide so, breeding for different kinds of rind pattern is important in making watermelon more attractive to consumers. Therefore, the results in this study will be useful to watermelon breeders in manipulating rind patterns to attract consumer's interest.

The authors’ results support their conclusions, and the analysis of the data are adequate for the overall objective of the study. The figures are clear, easy to read, and understand. However, the manuscript is poorly written. There are numerous grammatical and typing mistakes throughout the manuscript which makes it difficult to appreciate the results presented. The entire manuscript should be revised to correct the grammar and typing mistakes. Some of the sentences that should be fixed are listed below but, they are only a sample of the sentences that should be corrected.

Many sentences contain “had been identified to regulate” and “had been”. The authors should consider revising these sentences.

Line 40  - It had been reported that many different genes regulated tomato peel color and pattern.

Line 46 - Furthermore, methylation of TAGL1 promoter had been identified to regulate the formation of green stripe in tomato peel [5].

Line 64 - 65 - As an important appearance quality, watermelon rinds had many variations and were rich in diversity.

Line 62 -63 - All these make it possible to study the inheritance pattern of important agronomic traits and fine map the function genes.

Line 68 -69  - Besides of these, there are also some other special rind patterns, such as yellow spots, yellow belly, uneven rind and so on [21-23]. For watermelon breeders, different kinds of rind pattern has alway been the research hotspot.

Line 71 - In the pervious studies, the researches on watermelon rinds were focus on the inheritance and preliminary mapping.

Line 79 - In addition, some other special rind patterns were also been investigated.

Line 84-86 - With the reference genome sequences available for watermelon ‘97103’ and ‘Charleston Gray’, total 32,869 SSR markers distributed in watermelon whole genome were developed according to the ‘97103’ genome [28], which make it easily to fine mapping the function genes and study the regulatory mechanisms.

Line 128 - Thephenotypeof watermelon inbred line WT20, WCZ and F1.

Line 101-102 - The intermittent strip pattern of watermelon inbred line WT20 can be easily distinguished at 10 DAP, then the intermittent strip can be observed more and more clearly during the development of the fruit.

Line 218-220 - The χ2 test showed that the segregation ratio in F2 population and BC1 population were in line with 3:1 and 1:1 respectively (Table S1), indicated that watermelon intermittent strip was a qualitative trait and regulated by a single gene, which was named as ClIS (Citrullus lanatus Intermittent Strip).

Line 252 - Thecandidate gene was mapped between dCAPS-C and dCAPS-E markers on chromosome 6.

Author Response

In the manuscript entitled “Genetic mapping of a candidate gene ClIS controlling intermittent strip rind in watermelon” the authors generated a segregating F2 watermelon population by crossing an intermittent strip watermelon inbred line, WT20 and a green strip watermelon inbred line, WCZ and showed that the intermittent strip is controlled by the gene Citrullus lanatus Intermittent Strip (ClIS). Through genetic analysis the author’s demonstrated that watermelon intermittent strip is a qualitative trait under the regulation of ClIS, a single dominant gene. This finding of this study will be of particular importance to watermelon breeders. In selecting fruits, consumers are particularly attentive to quality and appearance. Features such as a unique rind pattern can be attractive to consumers. Watermelon is an important fruit which is grown and consume worldwide so, breeding for different kinds of rind pattern is important in making watermelon more attractive to consumers. Therefore, the results in this study will be useful to watermelon breeders in manipulating rind patterns to attract consumer's interest.

The authors’ results support their conclusions, and the analysis of the data are adequate for the overall objective of the study. The figures are clear, easy to read, and understand. However, the manuscript is poorly written. There are numerous grammatical and typing mistakes throughout the manuscript which makes it difficult to appreciate the results presented. The entire manuscript should be revised to correct the grammar and typing mistakes. Some of the sentences that should be fixed are listed below but, they are only a sample of the sentences that should be corrected.

Answer: Thank you very much for your comments and suggestions. We are very sorry for our poorly written. We have checked the grammar and typing mistakes of the manuscript. And the manuscript was also edited by a native English-speaking colleague Dr. Muhammad Jawad Umer. We think it is better now.

Many sentences contain “had been identified to regulate” and “had been”. The authors should consider revising these sentences.

Line 40  - It had been reported that many different genes regulated tomato peel color and pattern.

Answer: We have modified it to ‘It has been reported that many different genes regulated tomato peel color and pattern.’

Line 46 - Furthermore, methylation of TAGL1 promoter had been identified to regulate the formation of green stripe in tomato peel [5].

Answer: We have modified it to ‘Furthermore, methylation of TAGL1 promoter has been identified to regulate the formation of green stripe in tomato peel [5]’

Line 64 - 65 - As an important appearance quality, watermelon rinds had many variations and were rich in diversity.

Answer: We have modified it to ‘As an important appearance quality, watermelon rinds were rich in diversity.’

Line 62 -63 - All these make it possible to study the inheritance pattern of important agronomic traits and fine map the function genes.

Answer: We have modified it to ‘All these make it possible to study the inheritance pattern and functional genes of some important agronomic traits in watermelon.’

Line 68 -69  - Besides of these, there are also some other special rind patterns, such as yellow spots, yellow belly, uneven rind and so on [21-23]. For watermelon breeders, different kinds of rind pattern has alway been the research hotspot.

Answer: We have modified it to ‘In addition, there are some other special rind patterns, such as yellow spots, yellow belly, uneven rind [21-23]. Different kinds of rind pattern have always been the research hotspot of watermelon breeders.’

Line 71 - In the pervious studies, the researches on watermelon rinds were focus on the inheritance and preliminary mapping.

Answer: We have modified it to ‘In pervious studies, the researches on watermelon rinds focused on the inheritance and preliminary mapping.’

Line 79 - In addition, some other special rind patterns were also been investigated.

Answer: We have modified it to ‘In addition, some other special rind patterns have also been investigated.’

Line 84-86 - With the reference genome sequences available for watermelon ‘97103’ and ‘Charleston Gray’, total 32,869 SSR markers distributed in watermelon whole genome were developed according to the ‘97103’ genome [28], which make it easily to fine mapping the function genes and study the regulatory mechanisms.

Answer: We have modified it to ‘With the reference genome sequences available, total 32,869 SSR markers which distributed in watermelon whole genome were developed according to ‘97103’ genome [28], which makes it easily for fine mapping the function genes and studying the regulatory mechanisms.’

Line 128 - Thephenotypeof watermelon inbred line WT20, WCZ and F1.

Answer: We have modified it to ‘The phenotypes of watermelon inbred line WT20, WCZ and F1.’

Line 101-102 - The intermittent strip pattern of watermelon inbred line WT20 can be easily distinguished at 10 DAP, then the intermittent strip can be observed more and more clearly during the development of the fruit.

Answer: We have modified it to ‘The phenotypes of intermittent strip in watermelon inbred line WT20 can be easily distinguished at 10 DAP, and then it can be observed more and more obvious during later development of the fruit.’

Line 218-220 - The χ2 test showed that the segregation ratio in F2 population and BC1 population were in line with 3:1 and 1:1 respectively (Table S1), indicated that watermelon intermittent strip was a qualitative trait and regulated by a single gene, which was named as ClIS (Citrullus lanatus Intermittent Strip).

Answer: We have modified it to ‘The χ2 test showed that the segregation ratio in F2 population and BC1 population was in line with 3:1 and 1:1 respectively (Table S1), indicating that watermelon intermittent stripe was a qualitative trait and regulated by a single dominant gene, we named it ClIS (Citrullus lanatus Intermittent Strip).’

Line 252 - Thecandidate gene was mapped between dCAPS-C and dCAPS-E markers on chromosome 6.

Answer: We have modified it to ‘The candidate gene was located between dCAPS-C and dCAPS-E markers on chromosome 6.’

Reviewer 3 Report

The manuscript deals with the genetic mapping of a candidate gene for the intermittent strip rind in watermelon. While being of general interest as a gene that contributes to a unique fruit skin coloration pattern, whether this particular modification would be of any interest is questionable and needs to be better substantiated. 

The calculated value of X2 for the (WT20×WCZ F2) cross in Suppl. Table 1 is wrong and needs to be corrected. Luckily, this does not make the overall discussion incorrect, but such errors should be avoided as they may lead the authors to wrong conclusions about hypothesized gene action(s).

Authors have listed about a quarter of all references from their own previous studies. While the watermelon might not be the most studied crop, there are still enough other teams working on it to make such a large proportion of self-citations inappropriate, especially as they also include works on other crops, i.e., cucumber).

There are also technical issues that plague the entire text and need to be corrected, such as the printing of the large numbers (i.e. in Line 22), sequence of tenses (i.e. in Line 28), fused words (i.e. in Line 53), etc.

Author Response

The manuscript deals with the genetic mapping of a candidate gene for the intermittent strip rind in watermelon. While being of general interest as a gene that contributes to a unique fruit skin coloration pattern, whether this particular modification would be of any interest is questionable and needs to be better substantiated.

Answer: Thank you very much for your comments. Whether the intermittent strip watermelon would be of any interest, we think it is a groping process. Our laboratory has been breeding a new watermelon variety with intermittent stripes, and we also hope to see consumer feedback and evaluation.

The calculated value of X2 for the (WT20×WCZ F2) cross in Suppl. Table 1 is wrong and needs to be corrected. Luckily, this does not make the overall discussion incorrect, but such errors should be avoided as they may lead the authors to wrong conclusions about hypothesized gene action(s).

Answer: Thank you very much for your care and seriousness. We are very sorry my carelessness. We have modifed the value of X2 in the revised manuscripe and supplementary table, it is right now.

Authors have listed about a quarter of all references from their own previous studies. While the watermelon might not be the most studied crop, there are still enough other teams working on it to make such a large proportion of self-citations inappropriate, especially as they also include works on other crops, i.e., cucumber).

Answer: For the references, our lab has been committed to the researches on the fruit quality of cucurbitaceae crops. Therefore, we may refer some of our own researches during this study, we think it is reasonable, and these references we citied are related to our research content.

There are also technical issues that plague the entire text and need to be corrected, such as the printing of the large numbers (i.e. in Line 22), sequence of tenses (i.e. in Line 28), fused words (i.e. in Line 53), etc.

Answer: Thank you very much for your comments. We have checked the technical issues of the whole manuscript, we think the revised manuscript is better now.

Reviewer 4 Report

This is an  interesting work as regards its topic.  Some aspects  such as  clarity , grammatical and punctuation should be improved.

Line  45: change  ‘which ultimately lead to the formation’ with  ‘which ultimately leads to the formation’

Line  53: change  ‘many other transcriptionfactors’  with  ‘many other transcription factors’

Line  54: change  ‘which was involved in’  with ‘which were involved in’

Line  66: change  ‘so on; According’ with ‘so on. According’

Line  70: change  ‘rind pattern has alway been’ with  ‘rind pattern have always been’  

Line  71: change  ‘In the pervious studies ‘with ‘In the previous studies’

Line  76: change  ‘confirmed to control by a dominant gene‘ with ‘confirmed to  be controlled by a dominant gene ‘

Line  80: change  ‘were also been investigated’ with ‘were also investigated.’

Line  81: change  ‘confirmed to control by a single’ with ‘confirmed to be controlled by a single

Line  81: change  ‘gene; A single’ with  ’ gene. A single’

Line  86: change  ‘which make it easily to’ with ‘which makes it easily to’

Line  97: change  ‘identified to relate’ with ‘identified to be related’

Line  110: change  ‘will be not only helpful understanding the regulatory’ with ‘will be not only helpful in understanding the regulatory’.

Line  117: change  ‘While’ with ‘Moreover’

Line  135: change  ‘While’ with ‘Instead,’

Line  148: change  ‘WCZ; Then,’ with  ‘WCZ. Then,’

Line  150 : change  ‘mixed’ with ’by mixing’  

Line  154 : Clarify the meaning of  ‘which total 1066 individuals were contained.’

Line  203 : change ‘fruit. While’ with   fruit. Instead’

Line  204 : change ‘to fruit mature’ with  ‘to fruit maturity’

Line  206 : change ‘and were genetically stable.’ with  ‘and were genetically stable’  

Line  208 :  clarify ‘individuals were intermittent strip, which is same as WT20’

Line  209 :  clarify ‘strip are dominant over green stripin this study’

Line  219:  , ‘indicated‘ may be ‘indicating that’?

Line  228:  change ‘markers, nine distributed’ with ‘9 distributed’’

Line  230:  change ‘the end of watermelon’ with  ‘the end of the watermelon’

Line  243:  change  data between two parents,’ with  ‘data between the two parents,

Line  299:  change  ‘In order to analysis the’ with ‘In order to analyze the’

Line  360:  change ‘In previous study, with ‘In a previous study’

Author Response

Response:

Thank you very much for your comments. We have modified the grammatical and punctuation according to your suggestions, you can check them in the revised manuscript. And we also checked and edited the whole manuscript, we think the revised manuscript is better now.

Round 2

Reviewer 2 Report

The authors have adequately addressed all the issues pointed out in my earlier review.